# Using eZIS to Predict Progression from MCI to Dementia in Three Years

**DOI:** 10.3390/diagnostics14161780

**Published:** 2024-08-15

**Authors:** Ya-Tang Pai, Hiroshi Matsuda, Ming-Chyi Pai

**Affiliations:** 1National Cheng Kung University Hospital, Tainan 704, Taiwan; tompizza0@gmail.com; 2Department of Neurology, Chang Gung Memorial Hospital, Linkou Medical Center, Taoyuan 333, Taiwan; 3Department of Biofunctional Imaging, Fukushima Medical University, Fukushima 960-1295, Japan; mhiroshilab@gmail.com; 4Division of Behavioral Neurology, Department of Neurology, National Cheng Kung University Hospital, College of Medicine, National Cheng Kung University, Tainan 701, Taiwan; 5Alzheimer’s Disease Research Center, National Cheng Kung University Hospital, Tainan 704, Taiwan

**Keywords:** early Alzheimer’s disease, mild cognitive impairment due to Alzheimer’s disease, easy Z-score imaging system, prediction

## Abstract

(1) Background: Mild cognitive impairment (MCI) due to Alzheimer’s disease (AD) progresses to dementia at a higher annual rate, while other MCIs may remain stable or even improve over time. Discriminating progressive from non-progressive cases of MCI is crucial and challenging. (2) Methods: A retrospective study of individuals with MCI was conducted at a university hospital located in southern Taiwan. The researchers collected demographic data, comorbidities, the scores of cognitive tests, three easy Z-score imaging system (eZIS) indicators (severity, extent, and ratio), Fazekas scale scores, mesial temporal atrophy (MTA) scores, clinical outcomes including deterioration of Cognitive Abilities Screening Instrument, Mini-mental State Examination, Clinical Dementia Rating Sum of Box scores, and the conversion from MCI to dementia. Those who converted to dementia in three years and non-converters were compared by the three eZIS indicators to test the predictive utility, and the clinical outcomes were evaluated by regression and ROC curve analysis. (3) Results: The three eZIS indicators were significantly higher in the group of progressive MCI than in stable MCI. eZIS severity is positively correlated with a deterioration in the scores of the Cognitive Abilities Screening Instrument and Clinical Dementia Rating Sum of Box. eZIS severity is also positively correlated with conversion from MCI to dementia. The AUC for severity is 0.719, and the optimal cutoff value of severity for predicting conversion is 1.22. (4) Conclusions: During three years of follow-up, MCI individuals with greater eZIS severity were significantly associated with worse cognitive assessment scores and a higher conversion rate to dementia.

## 1. Introduction

The prevalence of mild cognitive impairment (MCI) without dementia among people over 71 is 22%. While 11.7% of the patients with all-cause MCI convert to dementia annually, prodromal Alzheimer’s disease (AD) progresses to dementia at a higher rate of 20.1% [1]. All in all, more than half of MCI cases progress to dementia within 5 years [1,2]. Some MCI patients, however, remain stationary or even improve in cognitive function over time. Clinically, to identify which MCIs will progress along the AD continuum is important, as cholinesterase inhibitor (ChEI) may have benefits to early AD patients [3]. Moreover, with new disease-modifying therapies (DMTs) having been recently approved [4,5], the discrimination of MCI due to AD from MCI due to non-AD is even more salient than before. Without using amyloid positron emission tomography (PET) scan, cerebrospinal fluid (CSF), or plasma biomarkers, it is difficult to differentiate one neurodegenerative disease from another at the MCI stage, see Table 1.

Single-photon emission computerized tomography (SPECT) is commonly used to diagnose AD since it is more acceptable than amyloid PET in availability and affordability. The easy Z-score imaging system (eZIS) is a quantitative method for detecting brain hypoperfusion, and it is more objective and reliable than visual interpretation by physicians. eZIS-assisted SPECT accurately detects hypoperfusion in the posterior cingulate cortex, precuneus, and inferior parietal cortex, which are cerebral regions commonly involved in early AD [6]. The three indicators of the eZIS (severity, extent, and ratio) can adequately discriminate very early AD from healthy controls [7,8,9]. Three indicators for characterizing perfusion decreases in patients with very early AD were determined [9,10]: First, the severity of hypoperfusion in specific areas showing hypoperfusion in very early AD from the mean positive Z-score in a specific volume of interest (VOI) consisting of the posterior cingulate gyrus, precuneus, and parietal lobes. Second, the extent of areas showing significant hypoperfusion in a specific VOI, i.e., the percentage of coordinates where the Z-score exceeds a threshold value of 2. Third, the ratio of the extent of areas showing significant hypoperfusion at a specific VOI to the extent of areas showing significant hypoperfusion in the whole brain. This ratio indicates the specificity of the hypoperfusion at a specific VOI relative to the hypoperfusion in the whole brain. When combined with Mini-mental State Examination (MMSE) scores, the Voxel-based Specific Regional Analysis System for Alzheimer’s Disease (VSRAD), and eZIS, the discrimination between MCI and early AD is more powerful than using MMSE alone [11]. Meanwhile, higher values of eZIS indicators of the posterior cingulate gyrus, precuneus, and parietal lobe have been reported to be associated with AD pathology [12,13].

In terms of the association between eZIS indicators and cerebral amyloid pathology, one study compared two groups of amnestic MCI patients. The results showed that a higher eZIS ratio was significantly related to positive 11C-Pittsburgh Compound B positron emission tomography (PiB-PET) in MCI [12]. Another study also revealed that among 37 probable AD dementia patients with an amnestic presentation according to the National Institute on Aging–Alzheimer’s Association criteria, higher eZIS severity, extent, and ratio were associated with the PiB-PET-positive group (24 patients) compared to the PiB-PET-negative group (13 patients) [13].

In differentiating between AD and non-AD patients, the eZIS also plays a significant role. eZIS severity was greater in AD than in vascular dementia (VD) and frontotemporal dementia (FTD) patients; eZIS extent was higher in AD than in FTD; and the eZIS ratio was greater in VD and FTD [14]. Additionally, higher eZIS severity, extent, and ratio were all associated with early-onset AD compared to late-onset AD [15].

Based on the aforementioned studies, the three eZIS indicators show a powerful capacity to discriminate dementia of AD from non-AD type [14], support the presence of cerebral amyloid pathology (CAP) in amnestic MCI (aMCI) individuals [12,13], and are correlated with the scores of mental examination [15]. When in the clinical stage of MCI, many patients and their families are eager to know whether the condition will advance with age. In this study, we used the eZIS to answer this question.

## 2. Participants and Methods

### 2.1. Subjects and Clinical Assessments

We conducted a retrospective study of MCI individuals who visited the memory clinic at the National Cheng Kung University Hospital (NCKUH) from January 2017 to December 2019. All patients on their first visit were over 50 and presenting with slowly progressive memory impairment, but not demented. Their history was confirmed by credible informants. Those with major depression or other psychiatric disorders and a history of traumatic brain injury or stroke were excluded. All underwent laboratory tests, including complete blood count (CBC), aspartate aminotransferase (AST), alanine aminotransferase (ALT), creatinine (Cr), blood urea nitrogen (BUN), thyroid stimulating hormone (TSH), thyroxine (T4), vitamin B12, folate, and venereal disease research laboratory tests (VDRL); neuroimaging: brain magnetic resonance imaging (MRI) and brain ECD SPECT; and cognitive function tests: Mini-mental State Examination (MMSE) [16], Cognitive Abilities Screening Instrument (CASI) [17], and Clinical Dementia Rating (CDR) Scale [18]. In addition, the Fazekas scale of deep white matter hyperintensity (DWMH) and periventricular white matter hyperintensity (PWMH) and Scheltens’ medial temporal atrophy (MTA) were obtained [19,20]. DWMH and PWMH were discriminated based on the commonly used “continuity to ventricle” rule [21]. The MTA score was graded from 0 (no atrophy) to 4 (severe atrophy) based on the width of choroid fissure, width of temporal horn, and height of hippocampal formation [20]. Both the Fazekas scale and MTA score were visually interpreted by author YTP.

### 2.2. Clinical Diagnosis

A diagnosis of MCI was made by a senior behavioral neurologist, MCP, according to Petersen’s criteria [22]: (1) memory complaint, preferably corroborated by an informant; (2) impaired memory function for age and education; (3) preserved general cognitive function; (4) intact activities of daily living; ND (5) not demented. Cognitive progression was assessed by deterioration in MMSE score [16], CDR sum of boxes (SB) [18], and CASI total score [17]. MCI conversion to dementia was defined by a global CDR of 1.0 or greater. The follow-up duration was at least 3 years. Those who progressed to dementia were categorized as progressive MCI. In contrast, patients who did not convert to dementia were categorized as stable MCI.

### 2.3. Data Analysis and Statistics

Independent variables included age of symptom onset, age when ECD SPECT exam was performed, education years, gender, comorbidities, ChEI administration and duration, interval between symptom onset and the performance of neuroimages, Fazekas scale of PWM and DWM changes, MTA, the three indicators of eZIS, and cognitive assessment scores (CASI, MMSE, and CDR SB). Dependent variables included deterioration of CASI, MMSE, and SB scores and conversion from MCI to dementia.

Student’s t-test and the chi-square test were used for continuous and categorical variables, respectively. Furthermore, we performed multiple regression analysis with stepwise selection to predict continuous outcomes (deterioration of cognitive function test scores) from independent variables; binary logistic regression with stepwise selection was performed to predict dichotomous outcomes (conversion from MCI to dementia) from independent variables. A two-tailed *p*-value < 0.05 was considered statistically significant in both univariate and multivariate analyses. In addition, the receiver operating characteristic (ROC) curve analysis was performed to evaluate the ability of the three eZIS indicators to predict MCI converting to dementia. An area under curve (AUC) > 0.7 was considered acceptable [23]. The Youden index was used to determine the optimal cut-off point [24].

## 3. Results

### 3.1. Clinical Features

We identified 86 MCI patients who met the research criteria throughout the study period, including 40 progressive MCI and 46 stable MCI, see Figure 1. No difference was detected between the two groups in age of symptom onset, age at the time of ECD SPECT exam performed, education years, gender, and comorbidities at the baseline. Regarding medication, more progressive MCI individuals were prescribed ChEI, while the duration of administration did not differ between the two groups. The interval between symptom onset and brain MRI exam and interval between symptom onset and brain SPECT exam did not differ between the groups. For the Fazekas scale, no difference was detected in DWM or PWM changes. For eZIS indicators, severity, extent, and ratio were significantly higher in the progressive MCI group, see Table 2.

### 3.2. Cognitive Assessments

For cognitive assessment scores at baseline, CDR-SB was higher in the progressive group, while no difference was present in CASI or MMSE, see Table 3. CASI, MMSE, and CDR-SB were significantly worse in the progressive group at the first, second, and third follow-ups. Again, no difference was found in the interval between baseline and the first follow-up, baseline and the second follow-up, or baseline and the third follow-up cognitive assessment.

Further analysis was performed to compare the nine CASI items between the two groups, including remote memory, recent memory, attention, mental manipulation, orientation, abstract, language, drawing, and verbal fluency, see Table 4. At baseline, recent memory and orientation were worse in the progressive group, while attention was better. At the first follow-up, recent memory and orientation were still worse in the progressive group, but attention showed no difference. At the second follow-up, in addition to recent memory and orientation, drawing and verbal fluency were also worse in the progressive group. At the third follow-up, six out of nine items were worse in the progressive group. When the baseline and the third follow-up scores were compared, a significant difference was noticed in all nine items between the two groups.

### 3.3. Outcomes

For outcomes, the deterioration of CASI, MMSE, and SB differed between the two groups, see Table 3. Furthermore, multiple regression was performed to predict deterioration of cognitive scores based on the variables including age of symptom onset, education, ChEI administration, ChEI duration, Fazekas scale of PWMH, Fazekas scale of DWMH, MTA, the three eZIS indicators, and baseline SB, see Table 5. The results revealed a positive correlation between CASI deterioration and MTA and eZIS severity, between MMSE deterioration and MTA, and between SB deterioration and MTA and eZIS severity. In addition, binary logistic regression was performed to predict the conversion from MCI to dementia based on the variables mentioned in multiple regression, see Table 6. The result revealed a positive correlation between the conversion and MTA and eZIS severity and baseline CDR-SB. A ROC curve was created to evaluate the ability of the eZIS indicators to predict the progression from MCI to dementia, see Table 7. The optimal cut-off values with the largest AUCs were 1.22 for severity (AUC = 0.719), 10.48 for extent (AUC = 0.712), and 1.01 for ratio (AUC = 0.698). To compare the predictive ability of MTA and eZIS indicators for the conversion from MCI to dementia, we conducted a further ROC analysis focusing on MTA. The optimal cut-off value, yielding the largest AUC, was found to be 2.5 for MTA (AUC = 0.720, sensitivity = 0.525, and specificity = 0.804), see Figure 2.

## 4. Discussion

In this 3-year follow-up study, MCI patients with an index of eZIS severity greater than 1.22 were more likely to convert to dementia. Although no amyloid PET scan or CSF/plasma biomarkers were examined, based on many previous reports, we can speculate that a large proportion of the progressive MCI patients in this study have CAP. First, MCI due to AD progresses to dementia at a higher annual rate compared to MCI due to non-AD [1]. In the same way, studies also showed that MCI with CAP were more likely to progress to dementia compared with those without [25,26]. In the Amsterdam Dementia Cohort and Subjective Cognitive Impairment Cohort (SCIENCe) project, according to the ATN classification system, subjective cognitive decline (SCD) individuals with CAP were at higher risk of progression to dementia or MCI compared with those without [27]. Secondly, the eZIS indicators have been shown to be positively associated with CAP [12,13]. Thirdly, a recent study that enrolled 67 MCI with CAP and 54 MCI without CAP and 64 controls also reported that during 5-year follow-up, MMSE and CASI deteriorated over time in MCI with CAP, while MCI without CAP and control subject did not exhibit an obvious trajectory of cognitive decline [28]. In sum, the three eZIS indicators could aid clinicians to make a diagnosis of MCI due to AD. Since early AD may experience benefits from ChEI, with the application of the eZIS to MCI patients, prescription of ChEI could be more accurate. Moreover, under certain conditions, the eZIS may be of help to pre-select candidates for new DMT for AD [4,5].

The deterioration rate in cognition and daily function over time may reflect the stage of MCI. MCI has been further categorized as early and late stage in several studies. In the Alzheimer ’s Disease Neuroimaging Initiative (ANDI), for example, MCI was classified into early and late based on WMS-R Logical Memory II story A score [29]. The ADNI reported that annual conversion rate from MCI to dementia was 2.3% for early MCI and 17.5% for late MCI [30]. A German study divided MCI into early and late based on delayed recall task of the Consortium to Establish a Registry for Alzheimer’s Disease (CERAD) battery [31]. Risk of conversion to AD was highest for late MCI compared with early MCI and controls. In our study, MCI was not further classified into early or late MCI. However, baseline CDR-SB was positively associated with conversion from MCI to dementia, which implied similar results to the mentioned studies. CDR-SB reflects functional ability and higher CDR-SB scores indicate more dysfunction or advancement in MCI stage and hence a higher likelihood to convert to dementia in three years. Nevertheless, more studies are needed to evaluate the application of the eZIS to predict the conversion to dementia in early and late MCI.

The MTA scale has been identified as a predictor of progression from MCI to dementia. One study reported that for every one-point increase in MTA score, the hazard ratio for MCI progressing to dementia increased by 1.5 during an average follow-up of 34 months [32]. Another study found that individuals with a mean MTA score greater than 2 were more than twice as likely to transition from MCI to dementia over a 3-year follow-up period [33]. As shown in Figure 2, the AUC for MTA closely approximated that for severity. However, it is important to note that a limitation of MTA is its subjectivity, whereas the eZIS provides a relatively objective assessment.

Limitations of this study are addressed here. First, drug prescription and compliance may affect the results. Even though more patients with progressive MCI used ChEI, which has been shown to be a DMT in AD, the overall rate of progression was even higher than the rate in the stable MCI group. We thus speculated that progressive MCI might include more individuals with CAP. Second, from clinical observation and longitudinal studies, some individuals with CAP, as confirmed by amyloid PET or CSF levels, may remain cognitively stable in the MCI AD stage for 10 years or longer. Differences in an individual’s genetic background, lifestyle, or environmental factors may affect the rate of progression in AD. In this study, however, we did not test ApoE genotypes and did not inquire into lifestyle or environmental factors, which should be added in future research. Furthermore, the results of this study cannot be applied to populations which are excluded from the research criteria. Hence, caution is advised when applying the results to real-world scenarios. Finally, the development of the eZIS used parameters based on Japanese norms, and the accuracy of the eZIS application to Taiwanese patients must be carefully interpreted.

## 5. Conclusions

During three years of follow-up, aMCI patients with greater eZIS severity were significantly associated with worsened cognitive assessment scores and a higher conversion rate from MCI to dementia. More experience with real-world scenarios is needed to know more about the value of the eZIS in clinical practice.

## Figures and Tables

**Figure 1 diagnostics-14-01780-f001:**
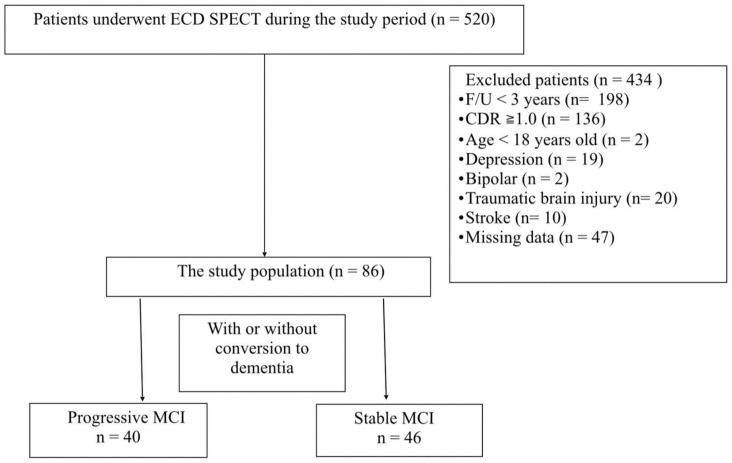
Patient selection flowchart.

**Figure 2 diagnostics-14-01780-f002:**
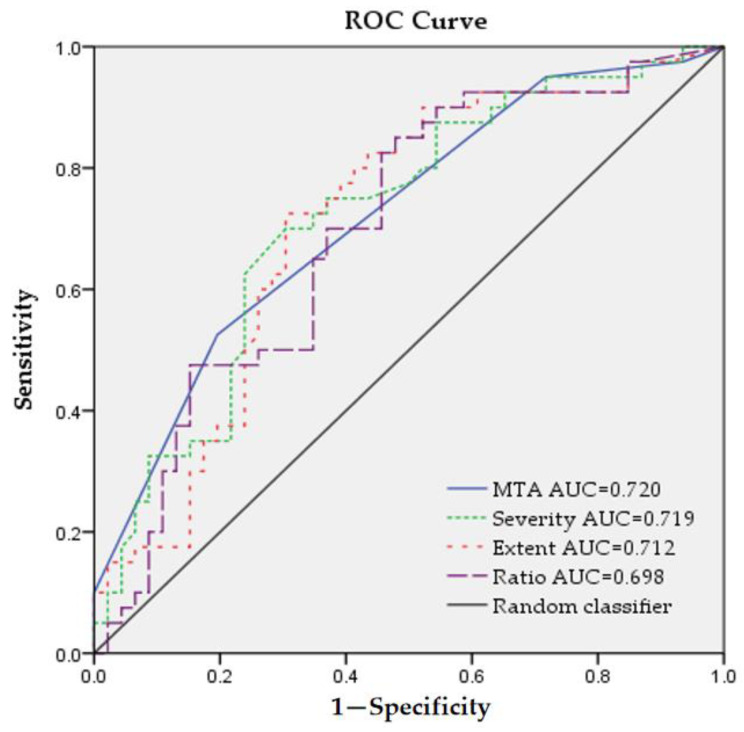
The receiver operating characteristic (ROC) curve of mesial temporal atrophy (MTA) and three eZIS indicators for predicting mild cognitive impairment (MCI) progressing to dementia. AUC: area under curve.

**Table 1 diagnostics-14-01780-t001:** Investigations for a diagnosis of early Alzheimer’s disease.

Category	Main Findings	Note
Clinical features	Impaired episodic memory, spatial navigation impairment	A, B
Neuropsychological tests	Long-term memory and orientation deficit	A, B
Neuroimaging, brain MRI	Mesial temporal atrophy	A
Neuroimaging, FDG PET scan	Hypometabolism in precuneus, posterior cingulate, inferior parietal lobes	C
Neuroimaging, brain SPECT	Perfusion reduction in precuneus, posterior cingulate, inferior parietal lobes	This study
Neuroimaging, cerebral amyloid or tau PET	Above threshold of centiloids	C, D
Fluid biomarkers	CSF or plasma levels of aß-40/42, p-tau and many new markers	D

Note: MRI: Magnetic Resonance Imaging; FDG: Fluorodeoxyglucose; PET: positron emission tomography; SPECT: single-photon emission computerized tomography. A: non-specific; B: posterior cortical atrophy syndrome and logopenic progressive aphasia, frontal variant of Alzheimer’s disease may have different profiles; C: high cost and low availability; D: only used in clinical trials or research currently.

**Table 2 diagnostics-14-01780-t002:** Demographic data and clinical characteristics of progressive MCI vs. stable MCI.

	Progressive MCI (n = 40)	Stable MCI (n = 46)	*p*-Value
Demographic data			
Age at onset, years (mean, [IQR], SD)	72.60 [66.00, 79.00] 8.82	71.00 [65.75, 77.00] 6.79	0.345
Age when SPECT performed (SD)	75.91 (8.64)	74.22 (6.31)	0.299
Education, years (mean, [IQR], SD)	8.55 [6.00, 12.00] 8.64	7.24 [5.25, 12.00] 4.76	0.218
Gender (male)	18 (45%)	17 (37%)	0.449
Comorbidities			
Hypertension	21 (52.5%)	23 (50%)	0.817
Dyslipidemia	6 (15%)	11 (23.9%)	0.301
Diabetes mellitus	6 (15%)	12 (26.1%)	0.207
Hearing impairment	11 (27.5%)	14 (30.4%)	0.765
Chronic kidney disease	6 (12.5%)	6 (13%)	0.940
Chronic heart failure	0 (0%)	0 (0%)	-
Coronary artery disease	6 (15%)	3 (6.5%)	0.200
Chronic obstructive pulmonary disease	2 (5%)	1 (2.2%)	0.476
Cancer	7 (17.5%)	7 (15.2%)	0.775
Cirrhosis	1 (2.5%)	0 (0%)	0.281
Hyperthyroidism	1 (2.5%)	1 (2.2%)	0.920
Hypothyroidism	3 (7.5%)	2 (4.3%)	0.533
Low vitamin B12	2 (5%)	2 (4.3%)	0.886
Low folic acid	2 (5%)	0 (0%)	0.125
Medication			
ChEI (prescription)	39 (97.5%)	31 (67.4%)	<0.01
ChEI duration, days (SD)	933.90 (438.12)	767.33 (697.38)	0.200
Intervals			
Symptom—MRI, days (SD)	1193.20 (725.02)	1166.43 (862.59)	0.878
Symptom—SPECT, days (SD)	1207.50 (800.55)	1174.43 (851.21)	0.854
Fazekas scale			
Deep white matter hyperintensity			
0–1	26	33	
2–3	14	13	0.501
Periventricular white matter hyperintensity			
0–1	16	26	
2–3	24	20	0.126
Medial temporal atrophy			
0–2	19	37	
3–4	21	9	<0.01
eZIS indicators			
Severity (mean, [IQR], SD)	1.42 [1.12, 1.68] 0.43	1.11 [0.84, 1.31] 0.39	<0.01
Extent (mean, [IQR], SD)	18.20 [8.98, 23.98] 13.24	9.61 [1.16, 16.53] 10.52	<0.01
Ratio (mean, [IQR], SD)	2.62 [1.37, 3.59] 2.11	1.59 [0.19, 2.25] 2.06	0.025

Note. IQR: interquartile range; SPECT: single-photon emission computerized tomography; ChEI: cholinesterase inhibitor; MRI: magnetic resonance imaging; eZIS: easy z-score imaging system; MMSE: Mini-mental State Examination; CDR: Clinical Dementia Rating; SB: sum of boxes; T0 was defined as the time when baseline cognitive function test was done; T1 was defined as the time when first follow-up cognitive function test was done; T2 and T3 were defined in a similar fashion; MCI: mild cognitive impairment.

**Table 3 diagnostics-14-01780-t003:** Cognitive assessments of progressive MCI vs. stable MCI.

	Progressive MCI (n = 40)	Stable MCI (n = 46)	*p*-Value
Cognitive function tests			
T0 CASI (mean, [IQR], SD)	69.83 [61.25, 77.75] 9.24	72.48 [66.75, 78.00] 8.59	0.171
T0 MMSE (mean, [IQR], SD)	20.50 [18.00, 23.00] 3.09	21.76 [19.75, 24.00] 2.85	0.052
T0 CDR, SB (mean, [IQR], SD)	1.81 [1.50, 2.50] 0.80	1.36 [1.00, 1.63] 0.58	<0.01
T1 CASI	64.55 [58.25, 75.50] 12.26	73.09 [67.75, 80.00] 8.66	<0.01
T1 MMSE	19.65 [18.00, 22.00] 4.44	21.50 [18.75, 24.00] 3.16	0.027
T1 CDR, SB	3.28 [2.00, 4.50] 1.86	1.77 [1.00, 2.50] 0.75	<0.01
T2 CASI	61.05 [54.00, 69.00] 13.04	73.52 [65.00, 82.25] 10.99	<0.01
T2 MMSE	18.53 [16.00, 21.75] 3.77	22.48 [21.00, 25.00] 3.38	<0.01
T2 CDR, SB	4.44 [3.00, 5.00] 2.17	1.86 [1.00, 2.63] 1.06	<0.01
T3 CASI	56.40 [52.00, 66.75] 16.12	73.87 [66.75, 81.25] 9.30	<0.01
T3 MMSE	16.45 [15.25, 19.00] 4.34	22.54 [20.00, 25.00] 3.05	<0.01
T3 SB	6.09 [4.50, 6.88] 2.70	1.86 [1.00, 2.50] 1.01	<0.01
Cognitive function tests follow-up interval			
T0–T1, days (SD)	434.58 (89.34)	418.74 (80.36)	0.389
T0–T2, days (SD)	809.05 (114.03)	790.57 (103.84)	0.434
T0–T3, days (SD)	1185.20 (148.22)	1167.13 (125.82)	0.543
Outcomes			
Delta CASI (T0–T3) (SD)	−13.43 (14.61)	1.39 (7.10)	<0.01
Delta MMSE (T0–T3) (SD)	−4.05 (4.25)	0.78 (2.74)	<0.01
Delta CDR, SB (T0–T3) (SD)	4.28 (2.80)	0.50 (1.15)	<0.01

Note. IQR: interquartile range; CASI: Cognitive Abilities Screening Instrument; MMSE: Mini-mental State Examination; CDR: Clinical Dementia Rating; SB: sum of boxes; T0 was defined as the time when baseline cognitive function test was done; T1 was defined as the time when first follow-up cognitive function test was done; T2 and T3 were defined in a similar fashion; MCI: mild cognitive impairment.

**Table 4 diagnostics-14-01780-t004:** Comparison of the CASI subitems between progressive and stable MCI.

	Progressive MCI (n = 40)	Stable MCI (n = 46)	*p*-Value
CASI items			
T0			
Remote memory	9.70 (0.72)	9.80 (0.58)	0.46
Recent memory	3.85 (2.53)	5.59 (2.46)	<0.01
Attention	7.35 (0.83)	6.87 (1.02)	0.02
Mental manipulation	7.43 (2.21)	6.65 (2.35)	0.12
Orientation	12.95 (3.83)	14.56 (3.34)	0.04
Abstract	6.50 (1.40)	6.26 (1.12)	0.38
Language	8.93 (1.19)	9.09 (0.98)	0.49
Drawing	8.18 (2.30)	8.04 (2.39)	0.80
Verbal fluency	4.93 (1.91)	5.67 (1.90)	0.07
T1			
Remote memory	9.57 (0.90)	9.61 (0.91)	0.86
Recent memory	3.43 (2.21)	5.80 (2.89)	<0.01
Attention	6.80 (1.20)	7.11 (0.92)	0.18
Mental manipulation	6.78 (2.67)	7.20 (2.28)	0.43
Orientation	10.80 (4.28)	13.85 (3.96)	<0.01
Abstract	6.15 (1.31)	6.35 (1.16)	0.46
Language	9.10 (0.98)	9.00 (0.94)	0.63
Drawing	7.35 (2.55)	8.17 (2.06)	0.10
Verbal fluency	5.10 (2.02)	5.89 (1.85)	0.06
T2			
Remote memory	9.25 (1.45)	9.74 (0.91)	0.06
Recent memory	3.13 (2.58)	6.28 (3.22)	<0.01
Attention	7.00 (0.96)	7.26 (0.85)	0.19
Mental manipulation	6.15 (2.98)	6.87 (2.29)	0.21
Orientation	9.00 (3.52)	13.65 (3.94)	<0.01
Abstract	6.53 (1.32)	6.52 (1.41)	0.99
Language	8.88 (1.26)	9.24 (0.77)	0.19
Drawing	6.88 (3.26)	8.24 (2.13)	0.03
Verbal fluency	4.25 (1.74)	6.20 (2.14)	<0.01
T3			
Remote memory	9.08 (2.04)	9.89 (0.43)	0.02
Recent memory	2.38 (2.27)	6.39 (3.00)	<0.01
Attention	6.58 (1.77)	7.11 (0.85)	0.09
Mental manipulation	6.53 (3.21)	7.30 (2.13)	0.20
Orientation	7.73 (3.18)	13.15 (3.90)	<0.01
Abstract	5.53 (2.18)	6.59 (1.11)	0.01
Language	8.20 (2.60)	9.28 (0.69)	0.01
Drawing	6.68 (3.32)	7.60 (2.51)	0.14
Verbal fluency	4.13 (2.14)	6.39 (1.87)	<0.01
Delta (T0–T3)			
Remote memory	−0.63 (1.90)	0.09 (0.63)	0.03
Recent memory	−1.48 (2.61)	0.80 (2.59)	<0.01
Attention	−0.78 (1.66)	0.24 (0.95)	<0.01
Mental manipulation	−0.90 (2.26)	0.65 (2.22)	<0.01
Orientation	−5.23 (3.89)	−1.41 (3.73)	<0.01
Abstract	−0.98 (1.91)	0.33 (1.23)	<0.01
Language	−0.73 (2.54)	0.20 (0.72)	0.03
Drawing	−1.50 (2.50)	−0.43 (1.90)	0.03
Verbal fluency	−0.80 (2.67)	0.72 (1.77)	<0.01

Note: data represent the mean (standard deviation); CASI: Cognitive Abilities Screening Instrument; T0 was defined as the time when baseline cognitive function test was done; T1 was defined as the time when first follow-up cognitive function test was done; T2 and T3 were defined in a similar fashion.

**Table 5 diagnostics-14-01780-t005:** Factors associated with cognitive function test score deterioration during three-year follow-up by multiple regression with stepwise selection.

	Conversion from MCI to Dementia	
Variables	Odds Ratio (95% CI)	*p*-Value
Age of onset	-	-
Education	-	-
ChEI use		
ChEI duration	-	-
PWMH	-	-
DWMH	-	-
MTA	3.05 (1.51–6.13)	<0.01
eZIS indicators		
Severity	5.35 (1.28–22.31)	<0.01
Extent	-	-
Ratio	-	-
T0 CDR, SB	3.08 (1.43–6.63)	<0.01

Note. ChEI: cholinesterase inhibitor; PWMH: periventricular white matter hyperintensity; DWMH: deep white matter hyperintensity; MTA: medial temporal atrophy; T0: baseline; CDR: Clinical Dementia Rating; SB = sum of boxes; MCI: mild cognitive impairment.

**Table 6 diagnostics-14-01780-t006:** Factors associated with the conversion from MCI to dementia during three-year follow-up by binary logistic regression.

	Delta CASI (T0–T3)		Delta MMSE (T0–T3)		Delta CDR SB (T0–T3)	
Variables	Coefficient	*p*-Value	Coefficient	*p*-Value	Coefficient	*p*-Value
Age of onset	-	-	-	-	-	-
Education	-	-	-	-	-	-
ChEI use	-	-	-	-	-	-
ChEI duration	-	-	-	-	-	-
PWMH	-	-	-	-	-	-
DWMH	-	-	-	-	-	-
MTA	−4.601	<0.01	−1.691	<0.01	1.007	<0.01
eZIS indicators						
Severity	−7.469	0.019	-	-	2.163	<0.01
Extent	-	-	-	-	-	-
Ratio	-	-	-	-	-	-
T0 CDR SB	-	-	-	-	-	-

Note. ChEI: cholinesterase inhibitor; PWMH: periventricular white matter hyperintensity; DWMH: deep white matter hyperintensity; MTA: medial temporal atrophy; T0: baseline; CDR: Clinical Dementia Rating; SB: sum of boxes; CASI: Cognitive Abilities Screening Instrument; MMSE: Mini-mental State Examination.

**Table 7 diagnostics-14-01780-t007:** The characteristics of the three eZIS indicators under ROC curve analysis to predict the conversion from MCI to dementia.

	AUC	*p* Value	Cut-Off Value	SEN	SPE
Severity	0.719	<0.01	1.22	0.700	0.696
Extent	0.712	<0.01	10.475	0.725	0.696
Ratio	0.698	<0.01	1.005	0.850	0.522

Note: AUC: area under curve; SEN: sensitivity; SPE: specificity.

## Data Availability

The data presented in this study are available on request from the corresponding author.

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
