# Peer review of "Using eZIS to Predict Progression from MCI to Dementia in Three Years"

_diagnostics, 2024, doi:10.3390/diagnostics14161780_

Round 1
Reviewer 1 Report
Comments and Suggestions for Authors
The manuscript evaluated the utilization of three eZIS indicators and other cognitive assessments in the prediction of MCI converting to dementia. This study is well designed, and the results are interesting, but there are several major concerns:
1. It would be very helpful if the authors introduce a little more about the eZIS indicators, since the most interesting finding of this study is the eZIS severity to predict progression from MCI to dementia.
2. The method needs to provide more information on how data about the eZIS indicators were achieved.
3. The presentation of the results is a little confusing. For example, Table 2 is the cognitive assessments, but it was included in “3.1 clinical features” instead of “3.2 cognitive assessments.” Line 188, There is no outcomes in table 4. Is it table 2? I would suggest the authors to reorganize the results.
4. It is necessary to discuss more about the utilization of three eZIS indicators, especially severity in previous studies to demonstrate the novelty or validity of this current study.
Comments on the Quality of English LanguageThe English is understandable, but there are several grammar mistakes in the manuscript.
Reviewer 2 Report
Comments and Suggestions for Authors
The authors of the manuscript describe an original clinical study aimed at solving an urgent scientific task – finding a method for effective diagnosis of early-stage Alzheimer's disease (AD), based on the analysis of three indicators of the easy Z-score Imaging System (eZIS) (severity, degree and ratio), which allows predicting the likelihood of AD development by the progression of moderate cognitive impairment (MCI).
Despite the limitations of the study (retrospective study design, small sample size, patient drug intake, specification of norms in eZIS criteria), the data obtained are of high importance for practical healthcare. The results shown by the authors can make a significant contribution to the development of methods for diagnosing and predicting the early stages of AD. Therefore, the conducted research may be relevant for publication in the journal Diagnostics.
The authors have developed an interesting and relevant study. The methods have been applied properly, the purpose and conclusions are clear. The manuscript is well written and detailed, and the research is well done.
As a small remark, I would like to note the need to reflect in the manuscript a table or figure with comparative characteristics of various available methods of early diagnosis of AD, indicating the place and scope of application for eZIS.
Round 2
Reviewer 1 Report
Comments and Suggestions for Authors
The revised paper has addressed all the comments.